# Descriptive Study of the Different Tools Used to Evaluate the Adherence to a Gluten-Free Diet in Celiac Disease Patients

**DOI:** 10.3390/nu10111777

**Published:** 2018-11-16

**Authors:** Luis Rodrigo, Isabel Pérez-Martinez, Eugenia Lauret-Braña, Adolfo Suárez-González

**Affiliations:** Gastroenterology Unit, Hospital Universitario Central de Asturias (HUCA), CSIC, Avda. de Roma s/n, 33011 Oviedo, Spain; ipermar_79@hotmail.com (I.P.-M.); meugelb@hotmail.com (E.L.-B.); adolfo.suarez@hcabuenes.es (A.S.-G.)

**Keywords:** celiac disease, gluten-free diet, effectiveness, adherence, nutritionists, clinic, serology, duodenal biopsies, structured questionnaires, peptides derived from gluten in faeces and urine

## Abstract

Celiac disease (CD) is a genetically conditioned autoimmune process that appears in susceptible people. It can affect people of any age, and slightly predominates in females. It has a fairly homogenous global distribution, with an average prevalence of 1–2%, the frequency having increased in recent decades. The only effective treatment is a strict and permanent gluten-free diet (GFD), although the level of compliance is poor, at about 50% of cases. To monitor the effectiveness of the GFD, several procedures involving various approaches are employed: (a) Periodic visits by expert Nutritionists; (b) Clinical follow-up; (c) Serological time controls of specific antibodies; (d) Serial endoscopies with collection of duodenal biopsies; (e) Use of structured questionnaires; and (f) Determination of gluten peptides derived from gluten in faeces and/or urine. All of these procedures are useful when applied, alone or in combination, depending on the cases. Some patients will only need to consult to their doctors, while others will require a multidisciplinary approach to assess their compliance with the GFD. In children, normalization of duodenal mucosa was achieved in 95% of cases within two years, while it is more delayed in adults, whose mucosa take longer time (3–5 years) to heal completely.

## 1. Introduction

Celiac disease (CD) is defined as a systemic autoimmune process that appears as a consequence of a permanent intolerance to gluten and that affects genetically predisposed people. It is widely, although unevenly, distributed throughout the world, affecting all ethnicities, with an average prevalence of 1–2% in the general population, its frequency having increased notably in recent years [1].

A gluten-free diet (GFD), followed strictly and permanently throughout life, is the only currently available treatment that successfully controls most cases of this disease. It is remarkably effective in the vast majority of cases, producing a significant clinical improvement not only in digestive symptoms, but also in the extra-intestinal symptoms associated with CD, causing its progressive disappearance, and the associated slow and sustained decrease in intestinal lesions [2,3].

Permanent adherence to a GFD is difficult, and largely unnoticed but repeated transgressions, as well as frequent contaminations may occur, all of which delay patient recovery in some way. This was confirmed in a meta-analysis published in 2018 [3]. If the transgressions are frequent, various types of associated long-term complications may appear, including a variety of malignancies [4].

Physicians involved in the management of CD should insist strongly to their patients that compliance with the GFD is fundamental and is the cornerstone of the success of this treatment. They need to explain this concept convincingly to the patients, as well as the main features of the GFD, with the greatest possible clarity and simplicity at the time of diagnosis.

It has not yet been well established which person or doctor who should carry out the follow-up to confirm the adherence to the GFD: it could be the gastroenterologist, the primary care physician, or an expert dietician [5]. Clinical control by family physicians and gastroenterologists is considered to be very similar, in the sense of them being able to achieve high rates of adherence to GFD [6].

The available evidence indicates that consulting with a dietician can be useful, especially when gluten contamination is suspected. However, joint monitoring by a dietician and a doctor may not be better, than the control that either one can offer separately [7]. The final decision will depend, not only on the availability of an expert dietician in different centres, but also on the collaboration that exists between the gastroenterology services and primary care physicians.

Patient associations or support groups can provide help with trying to achieve proper compliance with the diet. These associations offer detailed information about the importance of a strict GFD and answer all questions related to the characteristics of gluten-free foods and different recipes. They also organize regular meetings, during which patients can share information about CD with other patients and thereby improve compliance with their diet [8].

In general, it is reckoned that fewer than 50% of patients, mainly those from the adult celiac population, manage to keep strictly to a GFD. Generally, better dietary adherence is achieved (in 90–95% of cases, on average in the paediatric population, or in those people whose disease is diagnosed in early childhood [9].

There are clear differences in the prevalence of CD between Caucasians and South East Asian people, that may be due to different reasons, such as socioeconomic status, healthcare facilities, associated infections, presence of villous atrophy and others, and the adherence to GFD in general is irregular, but if it were strict, it would possibly be similar in both ethnicities, but for the moment, there are scarce information for this.

The main objective of the present study is to review the usefulness of the diverse available methods to use in different clinical situations and phases of CD evolution following a GFD, pointing out its strengths and weak points, in order to facilitate its selection, both in children and adults.

## 2. Control of the Follow-Up of the GFD

Compliance with the GFD can be evaluated through different approaches, and various health professionals, may participate or collaborate to carry this out, in line with the following study procedures:Periodic control visits by an expert dieticianRegular consultations with a gastroenterologist/family doctorStructured specific questionnairesRegular control of serum antibody titres for CDSerial endoscopies with duodenal biopsiesDetermination of derived peptides from gluten, in faeces/urine

### 2.1. Periodic Interviews Conducted by Dieticiass

The dieticians are the health professionals best placed to assess the degree of compliance with the GFD. They should try to participate and collaborate actively in the follow-up of celiac patients, whenever possible maintaining a close collaboration with nutritionists and gastroenterologists, ideally carrying out their task during the same visit and, as far as is practicable, simultaneously in the same consulting room.

However, many patients tend to have consultations only with their gastroenterologist or family doctor, and these are also effective at achieving strict adherence to the GFD. On the other hand, patients with more complex needs will require a multidisciplinary approach, including various medical specialists, to assess their associated diseases and their compliance with the GFD.

### 2.2. Evolutionary Clinical Follow-up

The disappearance or improvement of symptoms, might not seem a very accurate method, although an indicative one, for trying to evaluate adherence to the GFD during a consultation with the doctor. This evaluation can be done either by the gastroenterologist who diagnosed the patient, or by the family doctor who controls their evolution, facilitating their outpatient visits, reducing the number trips to the hospital and generally shortening the wait.

The lack of improvement with the GFD, or the persistence of symptoms, are generally related to irregular, or poor dietary compliance, or, clearly, with continued gluten consumption, specially in adolescents [10]. The main cause, in addition to any frequent transgressions, is the inadvertent intake of gluten. This is mainly due to “cross-contamination”, which arises from eating at a table with other people who eat bread or other wheat products and who often inadvertently drop small fragments of foods that have been contaminated with flour residues during their preparation or handling, etc. It may also be due to the “hidden gluten” that is present in some products that are not adequately labelled [11,12]. People with a basic elementary education and/or with little understanding of how to follow a GFD, or who have little motivation, often believe that they are following the diet strictly, although in practice, they are frequently making mistakes [13].

The follow-up and control visits serve to assess the improvement in the initial symptoms and to collect evidence of new ones. On many occasions at the time of diagnosis there is an overlap of the patient’s discomfort with the symptoms of irritable bowel, which, naturally, do not improve with the implementation of a GFD, unlike the case in many celiac patients [14]. The persistence of previous symptoms, or the appearance of new ones, may be related to CD itself, to a complication that has arisen, or to the presence of a new, associated disease.

A series of alarm symptoms have been described, such as the rapid deterioration in the general state of health, accompanied by striking weight loss, the presence of high fever, serious diarrhoea, or signs of intestinal obstruction, all of which require an exhaustive clinical and analytical study to be carried out in search of important associated complications, such as an evolution towards refractory forms of the disease, or the development of a primary intestinal T-cell lymphoma [15,16,17].

CD usually presents a broad spectrum of related symptoms, not only digestive, but also extra-intestinal. It has a clear predominance in females compared with men, with an average ratio of 2:1 in most series described. It is more prevalent in adults than in children. Children are more likely to exhibit typical symptoms along with a higher frequency of villous atrophy (VA) and higher antibody titres related to tissue Trans-Glutaminase (tTG) than in adults. Delay in diagnosis is also less in children than in adults in general. Curiously, this pattern is more evident in children under three years of age, and the differences between all other age groups decrease, or disappear with age, such that adolescents behave like adults in all the aspects discussed [18].

Children may have reduced bone mass at the time of diagnosis. However, they are more likely than adults to achieve a full recovery of their bone mass, 6–12 months after the starting on their GFD. Generally, it is not necessary to perform bone mineral density (BMD) explorations in paediatric patients who have been newly diagnosed with CD without complications. It is recommended to monitor the weight and height of children regularly, at every visit, in order to evaluate them properly, and to try to help them attain normal growth and skeletal development [19].

The main cause of a drop in BMD in CD patients is probably vitamin D deficiency, but it may arise for other reasons. However, the risk of fractures in patients with CD is not very high, and the predictive value of the systematic performance of BMD is not sufficient to identify individuals at high risk of fractures. It seems reasonable to measure BMD periodically in adult patients with CD, especially in those with a high risk of osteoporosis, such as in post-menopausal women, men older than 55 years, and people known to have osteopenia prior to the diagnosis of CD [20]. Further studies are needed to establish the true efficacy and cost-effectiveness of the periodic performance of BMD, in all adult patients with CD at the time of diagnosis, and to identify the frequency of follow-up with which this exploration should be carried out [21].

Autoimmune diseases are frequently associated with CD and can appear at any time during the follow-up, especially in adults, with a clear predominance in females. Physicians should be aware of the possible related autoimmune diseases and other illnesses associated with CD, so they can detect them systematically and early during follow-up visits [22].

CD is associated with many various diseases with which it often shares a genetic base. Type 1 Diabetes Mellitus (T1DM), is one disease, which occurs particularly frequently in children. Many of them present it in a silent or oligo-symptomatic way, for which reason, it is recommended to perform an annual follow-up screening for the presence of associated CD in children with T1DM, since its average prevalence is high, occurring in 5–10% of juvenile diabetic patients [23]. T1DM is diagnosed in 90% of patients before CD is confirmed [24]. Patients with T1DM and symptoms associated with CD, show clear clinical improvement overall when they follow a GFD. Increases in the rate of growth and in haemoglobin levels are often observed in these children. There is an improvement in control of diabetes mellitus, as confirmed by the reduction in the frequency of hypoglycaemic episodes and the reduced daily need for insulin [25,26,27].

There is also an increased risk of developing non-Hodgkin lymphoma in individuals with relatives affected with CD. The same may be said about the presence of various associated neurological diseases, above all cerebellar ataxia [28,29].

It is highly recommended and mandatory to carry out screening studies for CD in first and second grade relatives, especially if they present clinical symptoms. The index case must be informed about this family risk and its implementation is recommended for all first-and second-degree relatives [30,31].

The periodic determination of a series of laboratory tests is very important, in order to be able to detect the presence of nutritional deficiencies and the development of diseases and associated complications. The basic laboratory tests that should be done at each visit include, amongst others, a CBC with leukocyte formula, blood glucose, cholesterol and triglycerides, measurements of levels of iron, transferrin, ferritin, vitamin B12, folates, calcium, alkaline phosphatase, and liver function tests.

Likewise, the serum levels of thyroid stimulating hormone (TSH), anti-thyroid antibodies and levels of dihydroxy—vitamin D should be determined^,^ in case they exhibit an associated deficiency. This will be complemented with the determination of the antibody titre against deamidated gliadin peptides (DGPs) of the IgA type and/or the tTG, also of the IgA type, and occasionally by determining the endomysial antibodies (EMAs) [31].

Women of childbearing age must be checked regularly by their gynaecologist, especially those presenting menstrual disorders, infertility problems, or those who have a history of recurrent miscarriages [32].

Hypersplenism can affect more than a third of adult patients with CD, but it is not a complication in paediatric patients. The incidence of hyposplenism is correlated with the duration of pre-exposure to gluten and is higher in concomitant autoimmune disorders and premalignant conditions. Based on these associated factors, the function of the spleen can be determined in a select group of adult patients with CD and a previous history of associated major infections or episodes of thromboembolism [33].

The count of red blood cells with small surface irregularities (marks or notches) is a useful diagnostic tool, involving a precise, quantitative and low-cost method. The conjugated protein vaccines should be administered to patients with significant hyposplenism, defined as more than 10% of erythrocytes with irregularities on their surface or fewer than 10% of IgM memory type B cells [34].

### 2.3. Structured Questionnaires

Structured short questionnaires are used as an alternative to consultations with a dietician to obtain a rapid assessment of the adherence to the GFD. It is easy to complete this type of questionnaire in the patient’s usual clinic. The responses are highly correlated with antibody levels and the presence of VA in duodenal biopsies and useful for monitoring. In general, questionnaires are easy to administer and often complement each other. They not only assess the quality of life, in a general or specific way, but also are able to estimate the changes occurring after the follow-up of the GFD. However, all of them must be validated in different countries and diverse clinical contexts before they can be applied and come into general use. Leffler et al. have developed a simple questionnaire to assess adherence to GFD in adults with CD. It consists of seven structured questions about compliance and is scored on a Likert scale from 1 to 5, so that summing the values obtained gives an overall score from 7 to 35. Values less than 13 are considered to show good compliance, while those over 17, represent intermediate or low adherence. It is easily applied and is a very useful tool that can be included as part of the monitoring of adult patients, but not of children. This scoring system is known as “CDAT score” from (Coeliac Dietary Adherence Test) [35].

### 2.4. Periodic Evaluation of Serum Antibody Levels

Levels of antibodies circulating in the blood, which are usually used to diagnose CD, including DGPs and tTG, are related to the levels of gluten consumed. It is expected, therefore, that there will be a decrease in their titres a few weeks or months after a strict GFD is initiated. The sustained or sporadic consumption of foods containing gluten increases these values and thereby the persistence of high levels of gluten-related antibodies, suggests a lack of adherence to GFD [36].

Periodic serological tests for PDGs DGP and/or tTG may be useful for controlling the degree of compliance with a GFD [37]. However, the normalization of antibody titres cannot identify the existence of small dietary transgressions, so its use is limited to indicating a lack of compliance, but is of no value for evaluating whether there is strict adherence to the GFD.

The diagnostic security of the various commercial kits used is lower in clinical practice than the values reported in the medical literature in general, especially in the patients with mild VA or those with patchy lesions.

A collaborative multinational study revealed a high variability in levels of tTG, which is very striking with respect to diagnostic sensitivity (ranging from 69% to 93%), and somewhat less in terms of specificity (from 96% to 100%), among the 20 participating laboratories. This once again demonstrates the need for better standardization of the various techniques available for determining the tTG antibodies [38,39,40,41].

CD is often diagnosed in adults even when the values of positive antibody are very low (5–10% according to various series), or even zero, especially in cases that present with lymphocytic enteritis but not VA. The serology of such cases is entirely without value for monitoring because the levels of antibodies present at the start of the GFD are normal [42,43,44].

This does not usually occur in children, who, in most cases, have very high titres. Levels of tTG more than 10 times the normal value indicate the existence of significant VA. In these cases it is not considered necessary to perform duodenal biopsies in order to confirm the diagnosis of CD, as recommended in the new criteria drawn up by experts of the European Society of Paediatric Gastroenterology, Hepatology, and Nutrition (ESPGHAN), which were published in 2012 [45].

Subsequent follow-up studies have confirmed the safety and efficacy of this diagnostic strategy for children. It is recommended following a diagnosis based on raised levels of tTG, if it is confirmed by a positive determination of EMA in a second blood sample, and the presence of at least one symptom indicative of CD. This could avoid unnecessary risks and the costs related to endoscopy in at least 50% of the children worldwide who present with CD. Carrying out genetic studies to determine related HLA-II markers (HLA-DQ2 and DQ8) is not essential for obtaining a safe diagnosis [46].

The evaluation of antibody titres in these cases is a good, and indeed the only currently available method of monitoring. It is recommended that titres be measured every six months. Most children presenting with CD exhibit a gradual, progressive and continuous decrease in titres over time, their values becoming normalized within 1–2 years of starting the GFD.

### 2.5. Periodic Endoscopic Controls with Taking of Duodenal Biopsies

CD exhibits marked differences in children and adults, not only at the time of diagnosis, but also in the degree of response to the GFD and in their histological recovery. These are clearly higher in children (95% on average) than in adults (50%) after two years of monitoring [43,44].

Examining the histology of the small intestine continues to be the best procedure for evaluating with certainty the healing of the intestinal mucosa. Complete recovery from VA confirms that the GFD has been followed strictly and effectively, independently of the evolution of the serological titers of the antibodies, or of the symptoms that the patients present. Intestinal biopsies taken during follow-up are important in adults who have persistent VA, even in the absence of symptoms and with negative serology [47].

Adult patients are usually less symptomatic than children, most of them exhibiting atypical forms of the disease. For this reason, clinical-based monitoring is not usually of much value. Changes in serology are also not very marked, so the best way of assessing the effectiveness of the GFD is to perform periodic duodenal biopsies. These are not needed for children, because, in most cases, they achieve complete mucosal recovery earlier than do adults [48].

The guidelines published by the American College of Gastroenterology (ACG) include the recommendation to perform follow-up duodenal biopsies in adults, two years after the start of the GFD, in order to assess mucosal healing. However, its implementation is not recommended as a routine procedure for children [2]. Clinical guides published by the British Society of Gastroenterology are less demanding and suggest that there is little evidence to determine whether the clinical results differ significantly depending on the result of repeated biopsies. In the absence of results of a cost-benefit analysis of repeated biopsies and the lack of prospective studies in this regard, they postulate that follow-up biopsies are not generally necessary for asymptomatic patients who are successfully following a GFD and for whom there are no data to suggest an increased risk of developing VA-related complications [49].

One of the benefits of carrying out repeated biopsies in adult patients with CD is the ability to separate patients into two groups: those whose mucosa recovers completely and who can be monitored with less strict controls, and those with persistent VA, who require more frequent clinical control. It is clear that the persistence of VA is generally associated with a higher frequency of complications related to CD and adverse medium- and long-term outcomes.

Even patients who have persistent mild forms of lymphocytic enteropathy (Marsh I type or duodenal lymphocytosis) while following a GFD can also present nutritional deficiencies or complications. It takes 2–5 years for the mucosa of adult celiac patients with AV in their biopsies to recover. Therefore, the control biopsies can be programmed for adult patients over that period, taking into account that it is better to carry them out after the third or fourth year after initiation of the GFD to avoid too many unnecessary repetitions [50].

Mucosal recovery is defined, from the histological point of view, as the reestablishment of the normal height of the intestinal villi, taken as a ratio of villus height to crypt depth of at least 3:1 according to what is observed when comparing with non-celiac subjects. Monitoring the intra-epithelial lymphocyte count was not considered appropriate, because it is not exclusively related to gluten consumption, since it can also have other causes [51].

Usually the comparative evaluation of duodenal biopsies of the same patient, is mainly focused on the finding of the presence of an architectural damage (mainly VA). These parameters have been summarised following the Marsh-Oberhuber (MO) classification [52] and strongly influence to the gastroenterologists in their clinical/therapeutic decisions, especially when the histological pattern is unmodified [53].

Elli et al. reported the results of an interesting study including two different methods of comparing duodenal biopsies. One was the classical MO score and the second one compared the areas covered by each MO grade and expressed as percentages, the final grade being calculated from the analysis of ten power fields per duodenal biopsy. They studied 69 patients (17 males 52 females, age at diagnosis 36 ± 15 years) who underwent repeated duodenal biopsies. According to the classical MO scale, 32 patients (46%) did not present VA during one year follow-up, while 37 (54%) showed VA, among whom, 26 improved the grade of severity and 11 retained the same one. Of these latter, according to the second method, eight patients were considered improved, two showed a worsened duodenal damage and only one remained unchanged; the evaluation changed in 91% of cases. The authors suggest that the use of the second method provides a good additional information about the progression/regression of the mucosal damage, especially in unmodified cases following the GFD [54].

In 2010, Rubio-Tapia et al. described the results of a retrospective follow-up study with repeated biopsies in a large series of 241 adult patients with CD and VA at diagnosis. They reported that mucosal recovery in their patients was 34% (95% CI: 27–40%) at two years and 66% (95% CI: 58–74%) at five years. However, most of the patients (82%) presented a clinical improvement that was not significantly correlated with the recovery of the mucosa (*p* = 0.7), unlike the serological response, which did have a significant association (*p* = 0.01) In the same study, poor compliance with the GFD (*p* < 0.01), the presence of a severe CAD diagnosis, defined by the intensity of diarrhoea and weight loss (*p* < 0.001), and complete VA at the start (*p* < 0.001), were significantly correlated with a lack or delay of mucosal recovery. There was a nearly significant association between the disappearance of VA and a reduction in all-cause mortality (hazard ratio (HR) = 0.13, 95% CI: 0.02–1.06, *p* = 0.06), adjusted for age and sex. The authors recommended that most adults with CD should be followed up by endoscopy and periodic biopsies [55].

In another study, conducted by Lebwohl et al., published in 2013, the authors tried to establish whether there is an association between the presence of AV and increased mortality. They studied a series of 7648 patients with CD for a mean follow-up time of 11.5 years. A total of 3317 (43%) of them had persistent atrophy. There were 606 deaths (8%) in the entire series. Patients with persistent VA had no greater risk of mortality than those with normal mucosa (HR = 1.01; 95% CI: 0.86–1.19). This pattern was the same for children and adults, including patients older than 50 years of age. However, follow-up biopsies are useful for confirming diagnoses or in cases of refractory CD [56]. A description of the different published studies evaluating the adherence of the GFD and its impact on the duodenal mucosal healing and the survival in relation to persistence of VA is shown (Table 1).

The endoscopic capsule does not play any role in the follow-up of mucosal lesions in CD, since it does not allow biopsy samples to be taken. However, it is of use for detecting complications, especially when the appearance of an intestinal lymphoma or haemorrhages of unknown origin is suspected [57].

### 2.6. Determination of Peptides Derived from Gluten in Feces and Urine

Although the importance of controlling the follow-up of the GFD for effectiveness is clearly accepted by everyone, there are currently no unanimously accepted clinical guidelines that guarantee the results, nor are there adequate procedures to assess the adherence of the patients to the diet, and the transgressions that occasionally occur. Serological tests are very sensitive and specific to a diagnosis, but their effectiveness decreases with follow-up and they are not able to provide an adequate evaluation. It is difficult to ensure that repeated endoscopies are conducted, since they are inconvenient and invasive tests, so their application in practice is complicated. Recently, immunogenic gluten peptides (IGPs) have been identified that can be determined from faeces and urine. These have been proposed as simple, non-invasive markers, to be measured when frequent transgressions and/or contamination of the GFD are suspected. Their determination is simple and represents a new tool with which to determine compliance with the GFD objectively, at any time during the follow-up of patients with CD who are receiving treatment [58].

Several prospective studies have been carried out that have confirmed the efficacy of the determination of IGP in faeces. In a Spanish study of 188 patients with CD who were on a GFD, Comino et al. found that 56 patients (29.8%) had high levels of IGPs in their faeces, and that there were significant associations with age (39.2% over 13 years) and gender (a predominance of males in this evaluation). However, they found no correlation with antibody levels, or with the responses to the dietary questionnaires administered [59].

The determination of levels of IGPs in urine has been equally useful and simple as a way of monitoring the adherence of patients to the GFD. A highly effective lateral flow technique is used to detect the presence of the monoclonal antibody G12, which is the most immunogenic within the group of IGPs. This same procedure can be applied to the determination of the gluten content in food and in some beverages, such as beer [60,61].

## 3. Refractory Celiac Disease

Refractory celiac disease (RCD) is defined as the presence of persistent or recurrent symptoms of CD, accompanied by signs of malabsorption, with continued VA, despite the patient following a strict GFD for more than one year, in the absence of other complications, such as intestinal lymphoma. It is rare, affecting 1–2% of adult patients with CD and is divided into two categories: type I and type II [62,63].

In type I RCD, the lymphocyte infiltration of the mucosa of the small intestine is similar to that found in patients with CD before treatment with the GFD. In contrast, in type 2 RCD, the CD3 (+) intraepithelial lymphocytes present an abnormal immunophenotype, with a lack of expression of surface cells with CD8 (+) markers and that also feature oligoclonal or monoclonal type growth.

Their differences are not only immunophenotypic, but also involve their clinical behaviour, their response to medical treatment, and their prognosis. Type 1 RCD is the more frequent (60–80% of cases), and it responds better to treatment and has a better prognosis, while type 2 is the opposite, responding poorly to various treatments, being associated with a high frequency of intestinal lymphoma, and with greater mortality [64,65,66].

In a recent study of 57 patients who presented with RCD, in which immunosuppressive treatments had failed in more than half of the cases, oral treatment with budesonide in open capsules, dissolving their contents in apple juice at a dose of three 3-mg capsules per day, produced significant clinical improvement in 93% of cases, and histological improvement in 89% of cases, of both types [67].

Follow-up of both types of RCD should be frequent and complete, multidisciplinary, and based mainly on the findings of duodenal biopsies. These should be performed at regular intervals and with general examinations to rule out associated complications.

## 4. Indicative Timetable for Conducting the Evaluations

Once the diagnosis of CD has been confirmed and the treatment with GFD explained in depth by the gastroenterologist and the nutritionist, it is recommended to draw up a schedule of periodic review visits every six months during the first year, to evaluate the overall response to the GFD. This will be done from the clinical point of view, as well as from an analytical and serological perspective. In principle, it is not considered necessary to repeat the duodenal biopsies during the first year because, with a few exceptions, they are of little use.

After the first year, if the symptoms persist, the antibody titres remain elevated or if poor adherence to the GFD is confirmed, the possibility that the patient has RCD may be considered, in which case the surveillance and controls should be increased, applying the specific protocols of each hospital or the currently available clinical guidelines.

If this eventuality has been ruled out, and it is considered worthwhile to evaluate mucosal healing in adult patients, a control endoscopy can be arranged, with a biopsy taken after the third year of follow-up. In children, it is not considered necessary to perform provocation tests or duodenal control biopsies.

## 5. Conclusions

In clinical practice, CD that presents with digestive and/or extra-intestinal symptoms, but can also be oligo-symptomatic or occur in a silent way, is usually suspected from the finding of a positive serology and is confirmed by performing duodenal biopsies.

The effectiveness of compliance with the GSD by celiac patients can be monitored by a nutritionist or dietician through the repeated administration of questionnaires. However, these are not always available in all centres.

It must also be principally clinical, when the disappearance of the symptoms is evident, a significant improvement having been achieved. However, this is not sufficient for the other cases. Serological controls are only effective when the basal levels of the related antibodies are very high, but they are not useful when changes are small.

Carrying out periodic duodenal biopsies is useful but has the disadvantage of being an invasive procedure and, like serology, is more useful in cases with significant VA and that are not associated with clinical improvement or serological changes.

Finally, determination of the IGPs in isolated samples of faeces or urine has proved useful for controlling transgressions of the GFD.

However, some of these processes must be employed in order to monitor and encourage patient compliance and to rule out possible any associated shortcomings and/or the appearance of new complications.

## Figures and Tables

**Table 1 nutrients-10-01777-t001:** Summary of studies looking to GFD adherence and efficacy using various procedures.

Author, [Ref] City, Country Publication year	Number Type of pts.	Time of Follow-up	Study Procedures	Final Results
Dewar [16] London, England 2012	112 CD adults Non-responders to a GFD (NRCD)	18 months	Dietician Duod. Biopsies Colon biopsies H2-breath test	12, No CD 45%, not strict adherence GFD 11 MC; 9 SBBO; 9 RCD
Leffler [35] Boston, USA 2009	200 CD adults On GFD	Cohort evaluation	Dietician CDAT 7-item questionnaire Serology (tTG)	CDAT is easy to use and seems to be superior to periodic tTG determinations
Nachman [36] Buenos Aires, Argentina 2011	53 CD adults On GFD	At 1 and 4 years	Serum evaluation of tTG and DGP Compl. cut-offs	AUC at 1 year (0.64–0.72) AUC at 4 years (0.58–0.78)
Lebwohl [47] New York, USA 2014	7648 CD pts 3317 with VA 28 Swedish Pathology Departments	From 1969 to 2008 years Comparison of biopsies 2–5 years	Control of different predictor variables in VA persistent	VA is commoner in males (OR = 1.43)
Sharkey [48] Cambridge, England 2013	595 CD pts Paired biopsies in 391 cases Persistent VA in 47%	Retrospective study from database of only one hospital	Serum tTG sensitivity and VA persistent	Serology is a poor surrogate marker for the evaluation of mucosal recovery
Rubio-Tapia [55] Mayo Clinic, Minnesota, USA 2010	381 CD adults with biopsy proven CD	At 2 and 5 years	Clinical records Serological response Repeated biopsies VA persistent	Mucosal recovery was 34% (27–40%) at 2 years and 66% (58–74%) at 5 years (95% CI)
Lebwohl [56] New York, USA 2014	7648 CD pts 3317 with VA (43%)	Mean 11.5 years	Cox-regression Evaluating the mortality 606 pts (8%) in VA persistent	Persistent VA is not associated with increased mortality in CD

CD: celiac disease; GFD: gluten free diet; NRCD: non-responders to GFD; MC: microscopic colitis; SBBO: small bowel bacterial overgrowth; RCD: refractory celiac disease; CDAT: celiac dietary adherence test; tTG: tissue trans-glutaminase; DGP: deamidated gliadin peptides; Compl.: compliance; AUC: area under the curve; pts: patients; VA: villous atrophy.

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
