# Peer review of "Descriptive Study of the Different Tools Used to Evaluate the Adherence to a Gluten-Free Diet in Celiac Disease Patients"

_nutrients, 2018, doi:10.3390/nu10111777_

Reviewer 1 Report

To the editor and authors

The title of the paper is misleading: there is no evaluation described here, neither an efficacy study, but a summary of the approaches in order to evaluate the GFD adherence-this should be the title and the paper should be shortened and focused on this scope.

The English language and nomenclature should be extensively edited (e.g. line 127-BMD, D stands for density not "densitometry", line 145 various and not "varied" , line 165- a "complete hemogram"….that means  a CBC 5 diff, line 208 "PDG" means DGP ,231 "AEM" means EMA?  etc.).

The facts cited should be correct-for example line 155: the increased risk of developing Non Hodgkin Lymphoma is primarily in the CD affected individuals!

The paragraph lines 168-172: mixing between different tests celiac non specific (TSH and vitamin D) and specific (anti tTg, anti DGP, anti endomysial) without differentiation, meaning and not according to EPSGHAN guidelines.

Paragraph 182-186- not relevant to the topic.

Paragraph 2012-214- "Various commercial reagents"- meaning kits? In the next paragraph: tTg- very "marked"?

Line 235 "safe diagnosis" means an accurate diagnosis?

 Author Response

Response to Reviewer 1 Comments

 Point 1 : The title of the paper is misleading: there is no evaluation described here, neither an efficacy study, but a summary of the approaches in order to evaluate the GFD adherence-this should be the title and the paper should be shortened and focused on this scope.

Response 1: We fully agree with your suggestion of changing the title for ore more appropriate and  suggest this new one :

 “Descriptive study of the different tools used to evaluate the adherence to a Gluten-Free Diet in Celiac Disease Patients”

 We have changed this title in the manuscript (first page, lines 2-4)

Point 2 : The English language and nomenclature should be extensively edited (e.g. line 127-BMD, D stands for density not "densitometry", line 145 various and not "varied" , line 165- a "complete hemogram"….that means  a CBC 5 diff, line 208 "PDG" means DGP ,231 "AEM" means EMA?  etc.).

Response 2 :  OK .

 We accept all the suggestions and have incorporated all the changes into the text in the appropriated corresponding lines

Point 3 : The facts cited should be correct-for example line 155: the increased risk of developing Non Hodgkin Lymphoma is primarily in the CD affected individuals!

Response 3 :

Is in the line 186  not in the 155, where this fact is referred, we copy the results  of the study performed by Gao Y. et al. [28] and which the authors describe the results of a big epidemiologic study performed in Sweeden, giving the following results :

 “Overall they found persons with a hospital discharge diagnosis of CD to have a 5.35-fold (95% CI, 3.56-8.06) increased NHL risk. Risk of HL was borderline increased (OR=2.54, 95% CI, 0.99-6.56); however, there was no excess chronic lymphocytic leukemia risk.

Persons diagnosed with CD in 1975-1984, 1985-1994, and 1995-2004 had a 13.2-fold (95% CI, 3.63-48.0), 7.90-fold (95% CI, 3.38-18.5), and 3.84-fold (95% CI, 2.28-6.45) increased risk of NHL, respectively (P(trend)< .0001). Individuals with a sibling affected with CD had a 2.03-fold (95% CI, 1.29-3.19) increased NHL risk.”

 These are convincing results in the medical literature  in favour of the existence of an increased risk association between NHL and CD

Point 4 : The paragraph lines 168-172: mixing between different tests celiac non specific (TSH and vitamin D) and specific (anti tTg, anti DGP, anti endomysial) without differentiation, meaning and not according to EPSGHAN guidelines.

 Response 4 :  This is written in the paragraph 200-204 and textually says  as follow :

 Likewise, the serum levels of thyroid stimulating hormone (TSH), antithyroid antibodies and levels of dihydroxyvitamin D should be determined, in case they exhibit an associated deficiency. This will be complemented with the determination of the antibody titer against deamidated gliadin peptides (DGPs) of the IgA type and/or the tTG, also of the IgA type, and occasionally by determining the endomysial antibodies (EMAs) [31].

 So the determinations of TSH and vit. D serum levels would be indicated in the case of presence of  associated deficiency and of course the determination of specific serum antibodies  for CD  is clearly indicated (mainly DGP in children and tTG in adults)

 Point 5 : Paragraph 182-186- not relevant to the topic.

 Response 5 :

 We find that the paragraph placed at the level of 204-206 is very relevant.

 This is because it is a very well known fact that many women can start with clinical decompensation of its CD during their pregnancies and also they can present many gynecological  problems such as menses irregularity,  amenorrhea periods, infertility, increased abortions, premature deliveries, low birth children,  and so on……

 Furthermore, this paragraph is very short. It  only takes 2.5 lines and has one reference and we do not consider  the necessity  to remove it, because we consider relevant to include this short comment on this paragraph

 Point 6  :  Paragraph 212-214- "Various commercial reagents"- meaning kits? In the next paragraph: tTg- very "marked"?

Response 6 :

 We think that you are referring to the paragraph 248-250

 Instead of reagents, we have changed by “kits” into the text and instead of marked we have changed by “striking”

 Point 7 : Line 235 "safe diagnosis" means an accurate diagnosis?

 Response 7 :

 In the line 248, we have changed accuracy by “security” into the text

In the line 268,  we have changed AEM by “EMA” in the text

Many thanks for your interesting comments and suggestions in order to improve the manuscript quality

Reviewer 2 Report

The authors deal with a relevant and probably underestimated topic, which is about the clinical-serological tools able to assess and follow-up the efficacy and compliance to GFD in celiac patients. This has also crucial clinical implications in the management and follow-up of celiac disease (CD) and its complications.

Overall, the topic is nicely introduced and well organized. Therefore, I do not have specific major concerns about this paper, although few comments require attention.

1) Please clearly state the aim of the study.

2) A brief paragraph on the relationship between GFD adhrence and some CD-related complications should be included. For instance, neurological involvement is relatively common and clinical neurophysiology plays a crucial role in this context (Pennisi M, et al. Front Neurosci 2017). Moreover, some of these techniques, such as transcranial magnetic stimulation (TMS), are known for their role in the early detection and follow-up of CD-associated brain changes, even at the subclinical level (Lanza G, et al. Int J Mol Sci 2018). In particular, TMS studies in CD revealed valuable findings in the assessment and monitoring of these changes both in de novo patients (Pennisi G, et al. Plos One 2014) and in those after short (Bella R, et al. Plos One 2015) and long period of GFD (Pennisi M, et al. Plos One 2017).

3) An illustrative figure showing the timetable  for conducting the evaluations, as well as a table summarizing the different approaches, would be useful and improve the readibility of the manusript. 

Author Response

Response to Reviewer 2 Comments

Point 1: Please clearly state the aim of the study.

Response 1:  OK. 
A new paragraph has been included at  the end of the Introduction, covering the lines 92 to 95 and the complete text
is written down : “The main objective of the present study is to review the usefulness of the diverse available methods to use
in different clinical situations and phases of CD evolution following a GFD, pointing out its strength and weak points,
in order to facilitate its selection, both in children and adults”

Point 2: A brief paragraph on the relationship between GFD adhrence and some CD-related complications should be included. For instance, neurological involvement is relatively common and clinical neurophysiology plays a crucial role in this context (Pennisi M, et al. Front Neurosci 2017). Moreover, some of these techniques, such as transcranial magnetic stimulation (TMS), are known for their role in the early detection and follow-up of CD-associated brain changes, even at the subclinical level (Lanza G, et al. Int J Mol Sci 2018). In particular, TMS studies in CD revealed valuable findings in the assessment and monitoring of these changes both in de novo patients (Pennisi G, et al. Plos One 2014) and in those after short (Bella R, et al. Plos One 2015) and long period of GFD (Pennisi M, et al. Plos One 2017).

Response 2: There is not a clear confirmed relationship that a poor, or irregular GFD adherence,  that may play any influence in the onset of some CD related complications.

 So, we prefer to avoid the comment  on this controversial topic and express our gratitude to you for the interesting suggested references about the neurologic involvement and the new study methods that may clarify its presence in the clinic probably now, but much better in the near future.

 Point 3: An illustrative figure showing the timetable  for conducting the evaluations, as well as a table summarizing the different approaches, would be useful and improve the readibility of the manusript.

Response 3:

It is a very complex task, to try to summarize and illustrate in one schematic figure the timetable of performing the different evaluations or in one table summarizing the diverse approaches that have been described before.

So, we decline your kind invitation on these both aspects.

 Both aspects have been fully explained in the section 4 : Indicative timetable for conducting the evaluations (lines 408 to 423)

An also in the conclusions of the study in the section 5 (lines 427-444)

Furthermore there are some Clinical Guides and International Recommendations such as the ACG and the BSG, that are also referred in the manuscript.

 To your knowledge we have included the Table 1 with several works (7 in total) at the end of the manuscript showing the different ways of evaluation the adherence of the GFD and the findings that we include also here.

 Table 1 : Summary of studies  looking to GFD adherence and efficacy using various procedures

 Author, [Ref]

City, Country

Publication year

Number

Type of pts.

Time of

Follow-up

Study

Procedures

Final Results

Dewar DH[16]

London, England

2012

112 CD adults

Non-responders

to a GFD

(NRCD)

18 months

Dietician

Duod. biopsies

Colon biopsies

H2-breath test

12, No CD

45% , not strict adherence GFD

11 MC ; 9 SBBO ;

 9 RCD  

Leffler DA [35]

Boston, USA

2009

200 CD adults

On GFD

Cohort

evaluation

Dietician

CDAT

7-item questionnaire

Serology (tTG)

CDAT is easy to use and seems to be superior to   periodic tTG determinations

Nachman F [36]

Buenos Aires

Argentina

2011

53 CD adults

On GFD

At 1 and 4 years

Serum evaluation of tTG and DGP

Compl. cut-offs

AUC at 1 year

(0.64-0.72)

AUC at 4 years

(0.58-0.78)

Lebwohl B [47]

New York

USA

2014

7648 CD pts

3317 with VA

28 Swedish Pathology Departments

From 1969 to 2008 years

Comparison of biopsies 2-5 years

Control of different predictor variables in

VA persistent

VA is commoner in males (OR=1.43)

Sharkey LM [48]

Cambridge

England

2013

595 CD pts

Paired biopsies in 391 cases

Persistent VA in 47%

Retrospective study from  database of only one hospital

Serum tTG sensitivity and VA persistent

Serology is a poor surrogate marker for the evaluation   of mucosal recovery

Rubio-Tapia A

[52]

Mayo Clinic

Minnesota

USA

2010

381 CD adults with biopsy proven CD

At 2 and 5 years

Clinical records

Serological response

Repeated biopsies

VA persistent

Mucosal recovery was 34% (27-40%) at 2 years and 66%   (58-74%) at 5 years [95% CI]

Lebwohl B [53]

New York

USA

2014

7648 CD pts

3317 with VA (43%)

Mean 11.5 years

Cox-regression

Evaluating the mortality

606 pts (8%) in VA persistent

Persistent VA is not associated with   increased mortality in CD

 CD= Celiac Disease; GFD= Gluten Free Diet; NRCD= Non-responders to GFD; MC= Microscopic colitis; SBBO= Small Bowel Bacterial Overgrowth; RCD= Refractory Celiac Disease; CDAT=Celiac Dietary Adherence Test; tTG= Tissue Trans-Glutaminase; DGP= Deamidated Gliadin Peptides; Compl.=Compliance ; AUC= Area Under the Curve; pts=Patients; VA= Villous Atrophy;

Reviewer 3 Report

The review by Rodrigo et al provides a comprehensive evaluation of the efficacy and adherence to gluten free diet (GFD) in patients diagnosed with Celiac Disease (CD). The review is overall well written and structured. It covers the most important factors/approaches needed to evaluate how well CD patients maintain and respond to a GFD.

My minor comments can be found below:

It will be helpful if the authors provide a table summarizing the key studies and the examining factors associated with adherence to GFD in CD patients

The authors should consider including a section on the variation in outcomes of CD patients on GFD based on their ethnicity e.g. Caucasians v/s South east Asians

A small paragraph underlining the importance of "food labelling" and how it should be addressed by dietitians could be included in the section describing "cross-contamination" and "hidden gluten" (lines 99-105)

Lines 197-201: While mentioning the study by Leffler et al, the authors might want to mention the scoring system (CDAT score: Coeliac Dietary Adherence Test)

Overall, this review provides a good resource to have an overview on the outcome of CD patients based on their adherence to GFD.

Author Response

Response to Reviewer 3 Comments :

 Thee review by Rodrigo et al provides a comprehensive evaluation of the efficacy and adherence to gluten free diet (GFD) in patients diagnosed with Celiac Disease (CD). The review is overall well written and structured. It covers the most important factors/approaches needed to evaluate how well CD patients maintain and respond to a GFD.

My minor comments can be found below:

Point 1 :  It will be helpful if the authors provide a table summarizing the key studies and the examining factors associated with adherence to GFD in CD patients

Response 1 :  We have included tha Table 1, that contains this information and it has been placed at the end of the manuscript. We show a copy of it :

Table 1 : Summary of studies  looking to GFD adherence and efficacy using various procedures

 Author, [Ref]

City, Country

Publication year

Number

Type of pts.

Time of

Follow-up

Study

Procedures

Final Results

Dewar DH[16]

London, England

2012

112 CD adults

Non-responders

to a GFD

(NRCD)

18 months

Dietician

Duod. biopsies

Colon biopsies

H2-breath test

12, No CD

45% , not strict adherence GFD

11 MC ; 9 SBBO ;

 9 RCD    

Leffler DA [35]

Boston, USA

2009

200 CD adults

On GFD

Cohort

evaluation

Dietician

CDAT

7-item questionnaire

Serology (tTG)

CDAT is easy to use and   seems to be superior to periodic tTG determinations

Nachman F [36]

Buenos Aires

Argentina

2011

53 CD adults

On GFD

At 1 and 4 years

Serum evaluation of tTG and DGP

Compl. cut-offs

AUC at 1 year

(0.64-0.72)

AUC at 4 years

(0.58-0.78)

Lebwohl B [47]

New York

USA

2014

7648 CD pts

3317 with VA

28 Swedish Pathology   Departments

From 1969 to 2008 years

Comparison of biopsies   2-5 years

Control of different   predictor variables in

VA persistent

VA is commoner in males   (OR=1.43)

Sharkey LM [48]

Cambridge

England

2013

595 CD pts

Paired biopsies in 391 cases

Persistent VA in 47%

Retrospective study from  database   of only one hospital

Serum tTG sensitivity and VA persistent

Serology is a poor surrogate marker for the evaluation of mucosal recovery

Rubio-Tapia A

[52]

Mayo Clinic

Minnesota

USA

2010

381 CD adults with   biopsy proven CD

At 2 and 5 years

Clinical records

Serological response

Repeated biopsies

VA persistent

Mucosal recovery was 34%   (27-40%) at 2 years and 66% (58-74%) at 5 years [95% CI]

Lebwohl B [53]

New York

USA

2014

7648 CD pts

3317 with VA (43%)

Mean 11.5 years

Cox-regression

Evaluating the mortality

606 pts (8%) in VA persistent

Persistent VA is not associated with increased mortality in CD

 CD= Celiac Disease; GFD= Gluten Free Diet; NRCD= Non-responders to GFD; MC= Microscopic colitis; SBBO= Small Bowel Bacterial Overgrowth; RCD= Refractory Celiac Disease; CDAT=Celiac Dietary Adherence Test; tTG= Tissue Trans-Glutaminase; DGP= Deamidated Gliadin Peptides; Compl.=Compliance ; AUC= Area Under the Curve; pts=Patients; VA= Villous Atrophy;

Point 2 :  The authors should consider including a section on the variation in outcomes of CD patients on GFD based on their ethnicity e.g. Caucasians v/s South east Asians

Response 2 :

A new paragraph comment has been included at the end of the introduction section, lines 87 to 91.

There are clear differences in the prevalence of CD between Caucasians and South East Asians people, that may be due to different reasons such as socioeconomic status, healthcare facilities, associated infections, presence of villous atrophy and others, and the adherence to GFD in general is irregular, but if this were strict, possibly would be similar in both ethnicities, but for the moment there are scarce useful information on it.

Point 3 :   A small paragraph underlining the importance of "food labelling" and how it should be addressed by dietitians could be included in the section describing "cross-contamination" and "hidden gluten" (lines 99-105)

Response 3 :

We have added this smalll paragraph between the lines134-137, before the references [11,12]

It is recommended to fix only on  buying gluten-free foods, to be a membership of a local or regional coeliac society and to program regular periodic consultations with a dietitian, in order to achieve a better adherence to the GFD and  to improve the understanding of food labels [11,12].

Point 4 : Lines 197-201: While mentioning the study by Leffler et al, the authors might want to mention the scoring system (CDAT score: Coeliac Dietary Adherence Test)

Response 4 :

It has been included into the text into the lines 240-241 before the reference number [35].

This scoring system is known as “CDAT score” from (Coeliac Dietary Adherence Test) [35].

Thanks a lot for your kind suggested comments in order to improve the quality of our manuscript.